# Comprehensive Survey of Seismic Hazard at Geothermal Sites by a Meta-Analysis of the Underground Feedback Activation Parameter $a_{fb}$

**Arnaud Mignan** [1,2,*] **, Marco Broccardo** [3] **and Ziqi Wang** [4]

[1] Institute of Risk Analysis, Prediction and Management (Risks-X), Academy for Advanced Interdisciplinary Studies, Southern University of Science and Technology (SUSTech), Shenzhen 518055, China
[2] Department of Earth and Space Sciences, Southern University of Science and Technology (SUSTech), Shenzhen 518055, China
[3] Department of Civil, Environmental and Mechanical Engineering, University of Trento, 38123 Trento, Italy; marco.broccardo@unitn.it
[4] Department of Civil and Environmental Engineering, University of California, Berkeley, CA 94720, USA; ziqiwang@berkeley.edu
[*] Correspondence: mignana@sustech.edu.cn

**Abstract:** Global efforts to tame $CO_2$ emissions include the use of renewable energy sources, such as geo-energy harnessing. However, injecting pressurised fluids into the deep underground can induce earthquakes, hence converting $CO_2$-related risk into seismic risk. Induced seismicity hazard is characterised by the overall seismic activity $a_{fb}$ that is normalised by the injected fluid volume $V$ and the parameter $b$ of the Gutenberg–Richter law. The $(a_{fb}, b)$ set has so far been estimated for a dozen of reservoir stimulations, while at least 53 geothermal fluid stimulations are known to exist, based on our survey. Here, we mined the induced seismicity literature and were able to increase the number of estimates to 39 after calculating $a_{fb}$ from related published parameters and by imputing $b$ with its expectation where this parameter was missing ($0.65 \leq b \leq 2.9$, with mean 1.16). Our approach was a two-step procedure: we first reviewed the entire literature to identify seismic hazard information gaps and then did a meta-analysis to fill those gaps. We find that the mean and median $a_{fb}$ estimates slightly decrease from $a_{fb} \approx -2.2$ to $a_{fb} = -2.9$ and $-2.4$, respectively, and that the range of observations expands from $-4.2 \leq a_{fb} \leq 0.4$ to $-8.9 \leq a_{fb} \leq 0.4$, based on a comprehensive review unbiased towards high-seismicity experiments. Correcting for potential ambiguities in published parameters could further expand the range of possibilities but keep the mean and the median relatively close to original estimates, with $a_{fb} \approx -2.3$ and $-2.4$, respectively. In terms of the number of earthquakes induced (function of $10^{a_{fb}}$), our meta-analysis suggests that it is about half the number that could previously be inferred from published $a_{fb}$ estimates (i.e., half the seismic hazard). These results are hampered by high uncertainties, demonstrating the need to re-analyse past earthquake catalogues to remove any ambiguity and to systematically compute $a_{fb}$ in future geothermal projects to reduce uncertainty in induced seismicity hazard assessment. Such uncertainties are so far detrimental to the further development of the technology.

**Keywords:** seismic hazard; anthropogenic hazard; geo-energy; risk transfer; meta-analysis

## 1. Introduction

With increasing energy needs and the current push towards renewables, enhanced geothermal system (EGS) plants, which can be implanted theoretically anywhere, represent a promising complement to generate both electricity and heat [1,2]. The main challenge facing the EGS industry today is the risk of induced seismicity [3–6]. The same problem is faced by any geothermal project requiring underground stimulation, not only an EGS.

Injection-induced seismicity is due to complex thermo-hydro-mechanical-chemical (THMC) processes involving fault activation, pore pressure diffusion and other alterations

of the rock material [7,8]. Despite the apparent complexity of the physical processes involved and the heterogeneities of the underground, induced seismicity follows (in most of the cases) surprisingly simple empirical laws at a first level of analysis; in particular, a linear relationship between injected flow rate and induced seismicity rate, as well as a parabolic growth of the seismicity cloud over time during the injection. The laws have been verified at many locations and have been explained by both nonlinear poro-elasticity and geometric operations on an overpressure field [9–14].

Operators mitigate the induced seismicity risk by using so-called traffic light systems [13,15]. Despite the availability of such methods, induced seismicity still appears uncontrollable when projects are terminated due to the occurrence of unexpectedly large earthquakes [3,16], questioning their overall useability [17]. For this reason, it is crucial to understand the a priori seismic risk faced by populations, which is quantifiable via probabilistic seismic risk assessment [5,18,19]. The *a*- and *b*-values of the Gutenberg–Richter law $N(\geq m_0) = 10^{a - b m_0}$, where $N$ is the number of earthquakes above magnitude $m_0$ [20], are the main parameters characterising seismic hazard [21]. In the induced seismicity context, the *a*-value is normalised to the total injected volume $V$ (in m$^3$) so that the underground activation feedback parameter becomes $a_{fb} = a - \log_{10} V$, commonly referred to as the seismogenic index, $\Sigma$, in poro-elasticity parlance [10]. Despite its central role in hazard assessment, $a_{fb}$ is rarely evaluated.

Here, we explore the $\left( a_{fb}, b \right)$ parameter space from a comprehensive meta-analysis of the underground stimulation experiments which have taken place in view of geothermal energy harnessing. We then discuss our results within the context of the global energy transition, emphasising the need for uncertainty estimation and uncertainty reduction for improved decision making against the potentially new energy security risk that is anthropogenic seismicity.

## 2. Materials and Methods

### 2.1. Literature Mining

Our survey consisted in finding all underground stimulation experiments related to geothermal energy harnessing. We first used the lists of projects provided in existing reviews [22–26] and then complemented our research with an additional Google Scholar search for "injection induced seismicity geothermal". We obtained a comprehensive list of 53 fluid injections worldwide, which are described in Table A1, including matching references and relevant quotes from a total of 44 articles and reports. Note that shale gas fracking, wastewater disposal and geothermal circulation tests were not included in this study. Of the 53 experiments originally investigated, useful information could only be extracted for 39 of them. By "useful", we mean parameters from which $a_{fb}$ could be derived (see Section 2.2). Table 1 lists the 39 fluid injections, each with a site identifier, the site name, the injected volume $V$ [m$^3$], the mean flow rate $\dot{V}_{mean}$ [L/s], the number of earthquakes $N$ above minimum magnitude $m_0$, the maximum observed magnitude $m_{max}$ and, when already provided, $a_{fb}$ and $b$. We directly retrieved $\dot{V}_{mean}$ from texts or figures when available; otherwise, we calculated $\dot{V}_{mean} = \frac{V}{\Delta t}$ with $\Delta t$ the fluid injection duration (see Table A1 in the Appendix A for details).

Parameters $a_{fb}$ and $b$ are only available for 36% and 64% of the cases, respectively. We observe $a_{fb,mean} = -2.15$, $a_{fb,median} = -2.2$, $b_{mean} = 1.16$ and $b_{median} = 1.01$. While parameters are observed in the ranges $-4.2 \leq a_{fb} \leq 0.4$ and $0.65 \leq b \leq 2.9$, the $a_{fb}$ interval may be biased towards high estimates since large earthquake sequences are more likely to get analysed and statistical results published than stimulations devoid of earthquakes or that trigger very few of them [13]. Such a bias has already been mentioned for wastewater-disposal-induced seismicity [27]. Our meta-analysis aims at avoiding this potential hazard overestimation.

**Table 1.** Induced seismicity parameters available in the geo-energy literature for ($a_{fb}$,$b$) inference.

| ID [*] | Site | $V$ (m³) | $\dot{V}_{mean}$ (L/s) | $N \geq m_0$ | $m_0$ | $m_{max}$ | $a_{fb}$ | $b$ |
|---|---|---|---|---|---|---|---|---|
| 77ta | Torre Alfina | $4.2 \times 10^3$ | N/A | 177 | N/A | 3.0 | N/A | N/A |
| 78ce | Cesano | $2 \times 10^3$ | N/A | N/A | N/A | 2.0 | N/A | N/A |
| 83fh | Fenton Hill | $21 \times 10^3$ | 100 | 850 | −3 | 0 | N/A | N/A |
| 87lm | Le Mayet | $42.8 \times 10^3$ | ~14 | 107 | −2 | −1 | N/A | N/A |
| 88hi | Hijiori | $2 \times 10^3$ | ~100 | 65 | −4 | −1 | N/A | N/A |
| 89fj | Fjällbacka | 200 | N/A | N/A | N/A | −0.2 | N/A | N/A |
| 91og | Ogashi | $10.1 \times 10^3$ | ~11 | 1504 [†] | −2.0 [‡] | N/A | $-2.65 \pm 0.1$ | 0.74 |
| 92hi | Hijiori | $2.1 \times 10^3$ | N/A | 90 | −4 | 0 | N/A | N/A |
| 93og | Ogashi | $5.4 \times 10^3$ | ~15 | 762 [†] | −1.2 [‡] | N/A | $-3.2 \pm 0.3$ | 0.81 |
| 93sf | Soultz-sous-Forêts | $25.9 \times 10^3$ | ~19 | 9550 [†] | −1.0 [‡] | 1.9 | $-2.0 \pm 0.1$ | 1.38 |
| 94ktb | KTB | 86 | ~3 | 54 [†] | −1.3 [‡] | 1.2 | $-1.65 \pm 0.1$ | 0.93 |
| 95sf | Soultz-sous-Forêts | $28.5 \times 10^3$ | ~30 | 3950 [†] | −1.2 [‡] | N/A | $-3.8 \pm 0.1$ | 2.18 |
| 96sf | Soultz-sous-Forêts | $13.5 \times 10^3$ | ~78 | 3325 [†] | −1.2 [‡] | N/A | $-3.1 \pm 0.3$ | 1.77 |
| 00ktb | KTB | $4 \times 10^3$ | ~1 | 2799 | −1.2 | 1.1 | N/A | N/A |
| 00sf | Soultz-sous-Forêts | $23.4 \times 10^3$ | ~45 | 6405 [†] | 0.6 [‡] | 2.5 | $-0.5 \pm 0.1$ | 1.1 |
| 02bu | Bad Urach | $5.6 \times 10^3$ | N/A | 290 | N/A | 1.8 | N/A | N/A |
| 03be | Berlin | $300 \times 10^3$ | ~80 | 134 | −0.5 | 3.7 | N/A | N/A |
| 03ha | Habanero | $14.6 \times 10^3$ | ~19 | 2834 [†] | 0.0 [‡] | 3.7 | $-0.95 \pm 0.05$ | 0.75 |
| 03sf | Soultz-sous-Forêts | $33 \times 10^3$ | ~30 | 4728 | −0.9 | 2.9 | N/A | 0.83 |
| 04ktb | KTB | $64.1 \times 10^3$ | ~3 | 2405 [†] | −1.0 [‡] | N/A | $-4.2 \pm 0.3$ | 1.1 |
| 04sf | Soultz-sous-Forêts | $9.3 \times 10^3$ | 30 | 923 | −0.3 | 2.3 | N/A | 0.81 |
| 05ha | Habanero | $22.5 \times 10^3$ | ~20 | 16,017 | −1.2 | 2.9 | N/A | 0.83 |
| 05pa | Paralana | $3.1 \times 10^3$ | ~7 | 7085 | −0.6 | 2.5 | N/A | 1.32 |
| 05sf | Soultz-sous-Forêts | $12.3 \times 10^3$ | ~36 | 449 | −0.3 | 2.7 | N/A | 0.89 |
| 06ba | Basel | $10.8 \times 10^3$ | ~23 | 2313 | 1.0 | 3.4 | $0.4 \pm 0.1$ | 1.65 |
| 07ge | Geysers | $10.5 \times 10^6$ | ~50 | 1776 | 1.4 | 3.2 | N/A | 1.22 |
| 07gs | Gross Schönebeck | $13 \times 10^3$ | ~25 | 68 | −1.8 | −1.0 | N/A | N/A |
| 10jo | Jolokia | 380 | ~1 | 73 [†] | −1.4 | 1.0 | N/A | N/A |
| 11dp | Desert Peak | $\sim65 \times 10^3$ | ~33 | 18 | 0.0 | 0.7 | N/A | N/A |
| 12ha | Habanero | $34 \times 10^3$ | ~22 | 23,960 [†] | −1.6 | 3.0 | N/A | 1.01 |
| 12nb | Newberry | $40 \times 10^3$ | N/A | N/A | 0.2 | N/A | −2.8 | 0.8 |
| 13rr | Raft River | $341 \times 10^3$ | ~9 | 187 | −1.3 | 1.0 | N/A | N/A |
| 13ri | Rittershoffen | $3.2 \times 10^3$ | ~41 | 831 [†] | −1.4 | 0.9 | N/A | 1.16 |
| 13sg | St. Gallen | $1.2 \times 10^3$ | N/A | 349 | −1.1 | 3.5 | N/A | 0.83 |
| 14nb | Newberry | $9.5 \times 10^3$ | N/A | 398 | 0.1 | 2.26 | −1.6 | 1.0 |
| 15as | Äspö | 0.1 | ~0.05 | 196 | −4.2 | −3.5 | N/A | 2.9 |
| 16po | Pohang | $12.8 \times 10^3$ | N/A | 98 | −1.0 | 3.3 | −1.65 | 0.65 |
| 17gr | Grimsel | 1.4 | N/A | 65 | −4.3 [‡] | −2.5 | −2.4 | 1.03 |
| 18es | Espoo | $18.2 \times 10^3$ | ~4 | 43,882 | −0.6 | 1.9 | N/A | 1.3 |

[*] Successive stimulations at the same site within a year were averaged; [†] during injection only, otherwise injection + post-injection periods assumed; [‡] $m_0 = m_c$, otherwise $m_0 = m_{min}$ assumed.

We hierarchise the data into four categories based on parameter value availability: A: $\left(a_{fb}, b\right)$ (14), B: $\theta = \{N(\geq m_0), m_0, m_{max}, b\}$ (11), C: $\theta = \{N(\geq m_0), m_0, m_{max}\}$ (10) and D: $\theta = \{m_{max}\}$ (4). The injected volume $V$ is available for the 39 stimulations. In 14 additional cases (Table A1), information is insufficient to estimate $a_{fb}$. The way $a_{fb}$ is retrieved depends on the category, as described in the next section.

### 2.2. Meta-Analysis

Our meta-analysis consists in filling the gaps for the $a_{fb}$ and $b$ columns given in Table 1. Since the $b$-value can only be determined from earthquake catalogues, we can only impute missing values, here, using the mean $b_{mean} = 1.16$. Then, for both categories B and C, we can estimate

$$a_{fb} = \log_{10} N(\geq m_0) + bm_0 - \log_{10} V \tag{1}$$

[10]. For category D, we also use Equation (1) considering $N(\geq m_0) = 1$. For fixing missing $m_0$s, we use the following pragmatic approach. We examine from Table 1 the statistical trend between $m_{max}$ and $m_0$; then, we use linear regression to infer the expected value of $m_0$, given the observed $m_{max}$ (Figure 1). Specifically, we consider $m_0$ as a latent variable. When only one event is reported, it follows that $N(\geq m_0) = 1$ and, by definition (from Equation (1)), $m_0 \leq m_{max}$. Finally, we use the expected value of $m_0 = 0.62 m_{max} - 2.15$, which is the regression line in Figure 1. Given the data in Table 1, by construction $E[m_0|m_{max}] \leq m_{max}$. Observe that the point estimate approach might underestimate or overestimate $a_{fb}$ because the true $m_0$ is unknown but on average, it reflects the trend emerging from Table 1. An upper bound for $a_{fb}$ is given by $m_0 = m_{max}$ (Figure 1, green line). However, this is clearly an overestimation, as it represents the limit value of the outcome space of the latent variable $m_0$. Using this approach, we go from 14 published estimates of $a_{fb}$ to 39. The results are presented in Section 3.

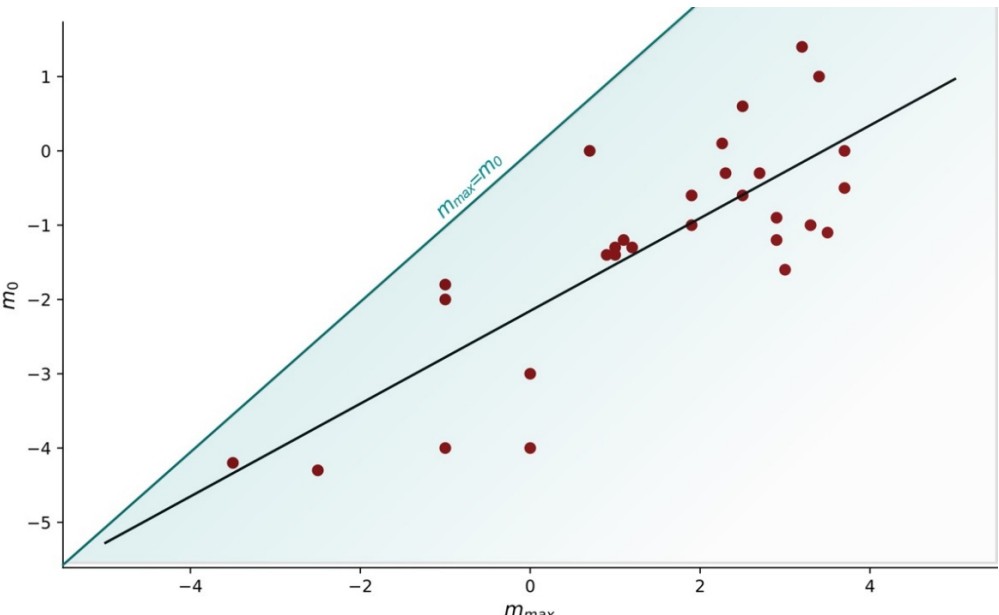

**Figure 1.** Statistical trend between $m_{max}$ and $m_0$ and matching linear regression $m_0 = 0.62\ m_{max} - 2.15$ in black.

It is important to note that some of the input data in Table 1 are ambiguous. Equation (1) is only valid for $m_0 \geq m_c$, with $m_c$ the completeness magnitude [28]. In most articles, we do not know whether this is the case or whether $m_0 = m_{min}$, the minimum observed magnitude. Moreover, $N$ possibly includes post-stimulation events in some of the articles. Although we cannot solve those two ambiguities from the literature alone, we can still

explore their possible impact on $a_{fb}$. If $m_0 = m_{min}$, $a_{fb}$ is underestimated because fewer events occur than predicted by the Gutenberg–Richter law if the data were complete at $m_0$. If $N$ includes post-stimulation events, $a_{fb}$ is overestimated by assuming more events being induced by the same volume $V$.

To investigate the $m_0$ ambiguity, we consider the magnitude–frequency distribution defined by Mignan [29], shown to also be valid for induced seismicity [26]:

$$\begin{cases} n_i(m) = 10^a 10^{(k-b)(m-m_c)} & , m < m_c \\ n_c(m) = 10^a 10^{-b(m-m_c)} & , m \geq m_c \end{cases}, \quad (2)$$

with $n_i$ and $n_c$ non-cumulative counts of earthquakes for the incomplete and complete magnitude ranges, respectively, and $k$ a detection parameter, found to be $\approx 3$ for both natural [29] and induced seismicity [26]. We obtain from Equation (2)

$$N = N_i + N_c = \int_{-\infty}^{m_c} n_i dm + \int_{m_c}^{+\infty} n_c dm = \frac{10^a}{(k-b)\log(10)} + \frac{10^a}{b\log(10)}, \quad (3)$$

with $N_i$ and $N_c$ the total counts of earthquakes for the incomplete and complete magnitude ranges, respectively. This leads to

$$N_c = \frac{N}{\frac{b}{k-b} + 1}, \quad (4)$$

To compute $a_{fb,corr} = \log_{10} N_c + bm_c - \log_{10} V$, we still need to estimate $m_c$. Assuming $n_i(m_0 = m_{min}) = 1$ and combining Equations (2) and (3) by substituting $10^a$, we get

$$m_c = m_0 + \frac{1}{k-b} \log_{10} \left( \frac{Nb(k-b)\log 10}{k} \right) \quad (5)$$

Finally, we obtain

$$a_{fb,corr} = \log_{10} \left[ \frac{N}{\frac{b}{k-b} + 1} \right] + b \left[ m_0 + \frac{1}{k-b} \log_{10} \left( \frac{Nb(k-b)\log 10}{k} \right) \right] - \log_{10} V, \quad (6)$$

Notice that if $m_0 = m_c$ (with $k \to \infty$ to collapse the incomplete magnitude range to zero, i.e., all events detected), we get back $N_c = N$ as well as Equation (1).

To investigate the post-injection ambiguity, we consider the induced seismicity model of Mignan et al. [13]:

$$n(t, \geq m_0) = \begin{cases} 10^{a_{fb} - bm_0} \dot{V}(t) & \text{for } t \leq t_s \\ 10^{a_{fb} - bm_0} \dot{V}(t_s) e^{-\left(\frac{t - t_s}{\tau}\right)} & \text{for } t > t_s \end{cases}, \quad (7)$$

with $\dot{V}(t_s)$ the injection flow rate at the shut-in time and $\tau$ the mean relaxation time. The first equation was first proposed in cumulative form by Shapiro's group [10] (Equation (1)). The linear relationship $n(t) \propto \dot{V}(t)$ has been confirmed by different theories and empirical studies [10–14]. The post-injection exponential decay has been validated on four induced seismicity sequences in the geothermal context with $\tau = (0.3, 1.1, 3.3, 12.6)$ days observed [13]. Fitted by a gamma distribution in Broccardo et al. [14], the most likely value of $\tau$ would be zero (i.e., $\tau_{mode} = 0$). Taking the integral of Equation (7) leads to

$$N = N_{inj} + N_{post} = 10^{a_{fb} - bm_0} \left[ V + \tau \dot{V}(t_s) \right], \quad (8)$$

with $N_{inj}$ and $N_{post}$ the number of earthquakes during injection and during the post-injection period, respectively [13]. Taking the average $\dot{V}(t_s) = \dot{V}_{mean}$ finally yields

$$a_{fb,corr} = \log_{10} N(\geq m_0) + bm_0 - \log_{10}\left(V + \tau\dot{V}_{mean}\right), \tag{9}$$

The potential impact of the two sources of ambiguity in the literature is discussed below.

## 3. Results

After applying the *b*-value mean point estimate as imputation of data categories C and D, we applied Equation (1) to calculate $a_{fb}$. The results are shown in Figure 2. The range expands from $-4.2 \leq a_{fb} \leq 0.4$ to $-8.9 \leq a_{fb} \leq 0.4$. Both the mean and the median decrease from $a_{fb,mean} = -2.15$ to $-2.91$ and from $a_{fb,med} = -2.2$ to $-2.4$, respectively. With the number of induced earthquakes being a function of $10^{a_{fb}}$, this suggests the expected number of induced earthquakes as being only 17% of what could previously be inferred from the literature, taking all other parameters constant. The downward effect is milder for the median, the number of induced earthquakes being 63% of what could previously be predicted. It is important to note that these results are subject to the errors associated with the application of Equation (1) and the imputation technique. Comparing category A (observed) to all categories combined (A + B + C + D) in Figure 2 shows the slight trend towards lower underground feedback activations. This is in agreement with the hypothesis that mainly high-seismicity experiments are statistically described in the scientific literature.

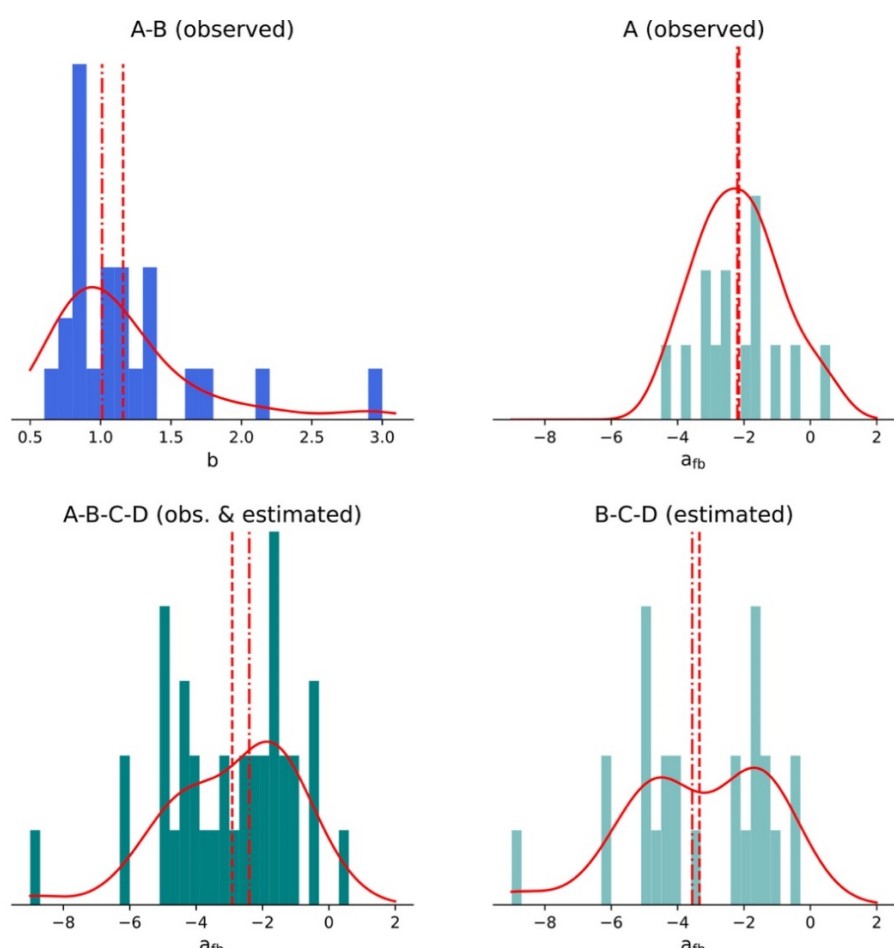

**Figure 2.** Empirical distributions of the seismic hazard parameters *b* and $a_{fb}$ derived from the meta-analysis of 39 fluid injections in the geothermal context. Gaussian density kernels are represented by red curves, mean estimates by red dotted lines and median estimates by red dashed lines.

We then investigated the potential role of ambiguity in the results. We used $\tau = 1$ day for the $N_{post}$ estimation. A comparison of the first row (original results without corrections) and the second row (with corrections for $m_0 = m_{min}$ and $N = N_{inj} + N_{post}$) of Figure 3 shows the expanded range $-8.9 \leq a_{fb} \leq 0.4$ further expanding to $-10.5 \leq a_{fb} \leq 3.1$, with corrected $a_{fb,mean} = -2.26$, while the median remains at $a_{fb,med} = -2.4$. Correcting for all ambiguous cases would suggest that the expected number of earthquakes is 78% of what could be inferred from published $a_{fb}$ estimates (still 63% when considering the median). However, it is not possible to know when corrections must be applied since the published information is ambiguous. Moreover, our equations Equations (6) and (9) are themselves approximative and only used as guides for the problem at hand. The potential biases of ambiguous inputs appear limited relative to the wide spread of $a_{fb}$ shown in the figures. The potential $a_{fb}$ decrease due to $N$ ambiguity could be dismissed if $\tau$ is confirmed to tend to zero [13]. The main problem is the potential $a_{fb}$ increase due to $m_0$ ambiguity. Combining all these results and the underlying assumptions, we can suggest that by considering the entire induced seismicity literature in the geothermal context, the average hazard is roughly half of what could previously be inferred.

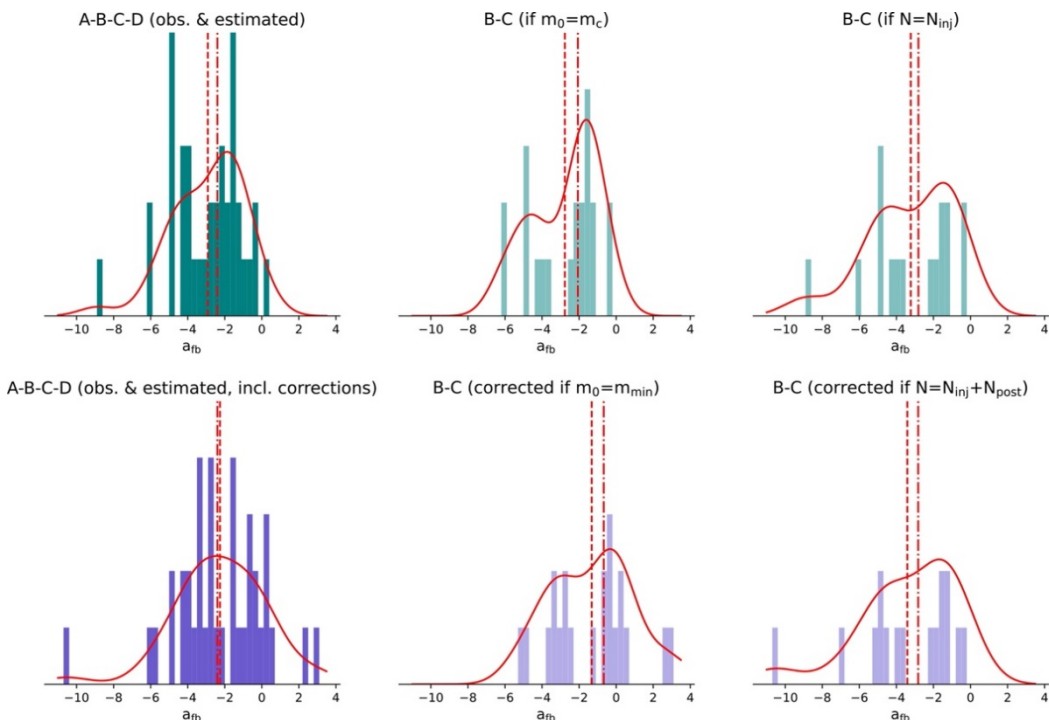

**Figure 3.** Empirical distributions of parameter $a_{fb}$ when investigating the ambiguity in input parameters $m_0$ and $N$. Gaussian density kernels are represented by red curves and median estimates by red dashed lines.

## 4. Discussion

It has been stated not only that underground fluid stimulation data are "too heterogeneous and too few in number to allow firm conclusions to be drawn on the basis of single-parameter correlation with seismic response" [23] but also that operational parameters are likely of secondary importance in explaining different induced seismicity behaviours [30]. Uncertainty and ambiguity lead to distrust of the public and to potential lack of entrepreneurship due to risk aversion [2]. By formalising induced seismicity hazard in terms of the two main parameters $a_{fb}$ and $b$ and by systematically assessing them by mining all available data from the literature (Table 1; Table A1), we made the first attempt at systematically assessing the seismic hazard in the geothermal energy harnessing context.

The parameter $a_{fb}$ is by nature highly uncertain, and any small change in parameterisation can have a significant impact on the $10^{a_{fb}}$ scale. Based on a comprehensive

meta-analysis of the geo-energy literature, we observed a widening of the range of possibilities and a slight decrease in the average $a_{fb}$ leading to a hazard about half of what could have been supposed before (yet, subject to data imputation and potential data ambiguity). In relative terms, the change may appear insignificant since the published range $-4.2 \leq a_{fb} \leq 0.4$ indicates a staggering 40,000-fold increase in the number of earthquakes, from the lowest underground activation feedback to the highest one observed in the world, for the same injected volume $V$.

The main message of our study is that reports of induced seismicity are often ambiguous. We recommend that researchers in this domain always provide $a_{fb}$ values as these are simple metrics to compute from any earthquake catalogue. The input parameters needed are simply the injected volume $V$ and the number of earthquakes $N_{inj}(\geq m_c)$ induced during injection above the completeness magnitude $m_c$. It would also be beneficial to re-analyse the earthquake catalogues of past fluid injections to remove any ambiguity in Table 1.

## 5. Conclusions

This work proves the importance of knowledge creation via data mining, which is complementary to expert elicitation [31]. The range of uncertainties is so vast in induced seismicity and the number of fluid injection experiments so scarce that it is difficult to comprehend the impact on decision making. Our actuarial approach, in contrast to expert judgement, provides the most accurate estimates of this hazard so far [32]. Ambiguities could easily be minimised by following the recommendations given in the Discussion section (Section 4). Ambiguity aversion increases risk aversion by emphasising the most extreme scenarios [33], such as what happened in Basel in 2006 (highest observed $a_{fb} = 0.4$) and in Pohang in 2017 (lowest observed $b = 0.65$). Projects were terminated in both cases [3,16]. High uncertainties still lead to risk aversion [2], but the range $\left(a_{fb}, b\right)$ being now quantified, the hazard can be compared to and ranked with other hazards in the energy security sphere.

**Author Contributions:** Conceptualisation, A.M.; methodology, A.M., M.B. and Z.W.; formal analysis, A.M.; investigation, A.M., M.B. and Z.W.; data curation, A.M.; writing—original draft preparation, A.M.; writing—review and editing, A.M., M.B. and Z.W.; visualisation, A.M. and M.B. All authors have read and agreed to the published version of the manuscript.

**Funding:** M.B. was supported by the Italian Ministry of Education, University and Research (MIUR), in the frame of the Departments of Excellence (grant L. 232/2016).

**Institutional Review Board Statement:** Not applicable.

**Informed Consent Statement:** Not applicable.

**Data Availability Statement:** All the data used in this study are listed in Tables 1 and A1.

**Acknowledgments:** We thank the two anonymous reviewers for their comments.

**Conflicts of Interest:** The authors declare no conflict of interest.

## Appendix A

We provide in Table A1 the raw data used to build Table 1, consisting of short quotes from 44 articles for 53 fluid injections. Of those, 14 cases do not provide enough information to estimate $a_{fb}$ but are mentioned for completeness and in the hope that acknowledging those gaps will help filling them in the future. Since the information retrieved from articles may be subject to interpretation, Table A1 provides all the details needed to reproduce Table 1 and to potentially modify it if corrections are ever needed.

**Table A1.** Raw literature excerpts used for data mining.

| ID | Site | Quotes |
|---|---|---|
| 74fh | Fenton Hill | NB: Most works about circulation and flow tests [34]. |
| 77ta | Torre Alfina | REINJECTION TEST [35] (p. 2); "*Vinj ($m^3$) = 4.2 × $10^3$*" [23] (Table 1); "*A total of 177 micro-shocks were recorded*" [35] (p. 3); "*maximum magnitude value of 3*" [35] (p. 3); [23] (Table 1). |
| 78ce | Cesano | INJECTION, "*Vinj ($m^3$) = 2 × $10^3$ ... Max $M_L$ = 2.0*" [23] (Table 1). |
| 79fa | Falkenberg | FRAC TEST [36] (p. 65); $V$ = 4.5 $m^3$ from "*0.2, 1.0 and 5.7 $m^3$ water were injected. The water loss during the third test was 2.4 $m^3$*" [36] (p. 67); "*(36 min pumping time for the 3 first experiments) a total of 60 seismic events were recorded on at least 2 seismic stations. 30 events could be used in the data analysis for source location*" [36] (p. 68). |
| 81la | Latera | REINJECTION TEST [35] (p. 2); $\Delta t \approx 50$ h, $\dot{V}_{mean} \approx 50$ $m^3$/h, $N$ for 1st test from [35] (Figure 3). |
| 83fh | Fenton Hill | HYDRAULIC STIMULATION, "*21,000 $m^3$ of water were injected*" [37] (p. 1); "*average flow rate of 0.1 cubic m/s ... Figure 4 shows only the 850 high quality events with magnitudes from −3 to 0*" [37] (p. 3). |
| 86hi | Hijiori | HYDRAULIC FRACTURING, "*A total of 1000 $m^3$ of water was injected*" [38] (p. 173). |
| 87lm | Le Mayet | INJECTION TESTS, "*total injected volume of 11,665 $m^3$ ... total injected volume for this phase reached 14 790 $m^3$ ... this phase reached 16,310 $m^3$ ... (11 events, 50 events, 46 events) associated with the large-scale injection tests ... the magnitudes of these events range between −2 and −1*", $\dot{V} \approx 8 - 20$ L/s [39] (p. 681). |
| 88hi | Hijiori | REINJECTION [38] (p. 176); HYDRAULIC FRACTURING, "*carried out from July 19 to 20 ... a total of about 2000 $m^3$ of water was injected*" [38] (p. 173); $\dot{V}_{mean} \approx 6$ $m^3$/min [38] (Figure 3); "*the hypoleft and magnitude of 65 microseismic events were determined. The largest event had a magnitude of −1.0*" [38] (p. 177); $m_0 = -4$ [38] (Figure 6); NB: Events during injection + post-injection [38] (Figure 3). |
| 88yu | Yunomori | "*Lack of data in the literature*" [25] (Table 1). |
| 89fj | Fjällbacka | STIMULATION, "*Vinj ($m^3$) = 200 ... Max $M_L$ = −0.2*" [23] (Table 1). |
| 91og | Ogashi | INJECTION, "*Time = 11 days, Volume ($m^3$) = 10,100, Event number = 1504, Mi = −2.0, b = 0.74, $\Sigma$ = −2.65 ± 0.1*" [10] (Table 1). |
| 92hi | Hijiori | HYDRAULIC FRACTURING, $N \approx 90$, $m_0 = -4$ and $m_{max} = 0$ [40] (Figure 7); "*Over 2100 t of water were injected*" [41] (p. 2). |
| 93co | Coso | STIMULATION, "*The total volume injected for the entire stimulation was only 12,700 $m^3$ (80,000 bbl). Significant microseismicity was recorded during the stimulation experiment*" [42]. |
| 93og | Ogashi | INJECTION, "*Time = 16 days, Volume ($m^3$) = 20,700, Event number = 762, Mi = −1.2, b = 0.81, $\Sigma$ = −3.2 ± 0.3*" [10] (Table 1). |
| 93sf | Soultz-sous-Forêts | STIMULATION, "*Max $M_L$ = 1.9*" [23] (Table 1); "*Time = 16 days, Volume ($m^3$) = 25,900, Event number = 9550, Mi = −1.0, b = 1.38, $\Sigma$ = −2.0 ± 0.1*" [10] (Table 1). |
| 94ktb | KTB | INJECTION TEST, "*Max $M_L$ = 1.2*" [23] (Table 1); "*Time = 9 h, Volume ($m^3$) = 86, Event number = 54, Mi = −1.3, b = 0.93, $\Sigma$ = −1.65 ± 0.1*" [10] (Table 1). |
| 95sf | Soultz-sous-Forêts | INJECTION, "*Time = 11 days, Volume ($m^3$) = 28,500, Event number = 3950, Mi = −1.2, b = 2.18, $\Sigma$ = −3.8 ± 0.1*" [10] (Table 1). |
| 96sf | Soultz-sous-Forêts | INJECTION, "*Time = 48 h, Volume ($m^3$) = 13,500, Event number = 3325, Mi = −1.2, b = 1.77, $\Sigma$ = −3.1 ± 0.3*" [10] (Table 1). |
| 00ktb | KTB | INJECTION, "*60-day, long-term fluid-injection experiment ... About 4000 $m^3$ of water were injected ... Of a total of 2799 induced events, hypoleft locations were obtained for 237 events*" [43] (p. 2369); "*The 125 events for which fault plane solutions were determined were found to cover a magnitude range of $-1.2 \leq M_w \leq +1.1$*" [44] (p. 5); "*the seismic network detected 2799 events ($-1.2 < ML < 1.1$)*" [45] (p. 997). NB: Inconsistent estimates; the Kwiatek et al. [45] information seems the less ambiguous one. |
| 00sf | Soultz-sous-Forêts | INJECTION, "*Time = 6 days, Volume ($m^3$) = 23,400, Event number = 6405, Mi = 0.6, b = 1.1, $\Sigma$ = −0.5 ± 0.1*" [10] (Table 1); $m_{max} = 2.5$ [46] (Figure 5). |
| 02bu | Bad Urach | STIMULATION, "*Vinj ($m^3$) = 5.6 × $10^3$ ... Max $M_L$ = 1.8*" [23] (Table 1); "*Out of 420 events monitored, 290 were located*" [47] (p. 874). |

**Table A1.** *Cont.*

| ID | Site | Quotes |
|---|---|---|
| 03be | Berlin | HYDRAULIC STIMULATION, "*581 events with moment magnitudes ranging between −0.5 and 3.7*" [48] (p. 98); "*The initially provided hypoleft catalog contained 581 events recorded between October 2002 and February 2004. During the stimulation periods, the seismic acquisition system recorded 134 events*" [48] (p. 100); $\dot{V}_{mean} \approx 80$ L/s [48] (Figure 2); $V = 300 \times 10^6$ litres [15] (Figure 10). NB: "*a magnitude 4.4. event . . . two weeks after shut-in*" [22] (p. 206). |
| 03bu | Bad Urach | INJECTION TEST, "*a total volume 3200 m³ of fresh water was injected. Induced seismicity during this experiment (218 events) was located*" [47] (p. 875). |
| 03ha | Habanero | INJECTION, "*Time = 9 days, Volume (m³) = 14,600, Event number = 2834, Mi = 0.0, b = 0.75, $\Sigma = -0.95 \pm 0.05$*" [10] (Table 1); "*A dozen of the strongest events . . . were assigned magnitudes between 2.5 and 3.7*" [49] (p. 2243). |
| 03ho | Horstberg | STIMULATION, "*Vinj (m³) = 20 × 10³ . . . Max $M_L$ < 0*" [23] (Table 1). |
| 03sf | Soultz-sous-Forêts | STIMULATION, "*we have selected 4728 events detected by the seismic network which have magnitude ranging from −0.9 to 2.9 . . . over 33 000 m³ of fluids were injected in GPK3,*" $\dot{V}_{mean} \approx 30$ L/s [50] (p. 1120); b = 0.83 [26] (Table 2). |
| 04ktb | KTB | INJECTION, "*Time = 223 days, Volume (m³) = 64,130, Event number = 2405, Mi = −1.0, b = 1.1, $\Sigma = -4.2 \pm 0.3$*" [10] (Table 1). |
| 04sf | Soultz-sous-Forêts | INJECTION, "*about 1250 events were detected . . . among them we selected 923 events . . . The magnitude of the events ranges from −0.3 to 2.3*" [51] (p. 51); "*lasted 3.5 days. The injection was maintained at a constant flow rate of 30 L/s . . . During this stage 9300 m³ of fluid were injected*" [51] (p. 52); b = 0.81 [26] (Table 2). |
| 05ha | Habanero | HYDRAULIC RESTIMULATION, "*continuing for 13 days, the Habanero 1 well was restimulated by injecting a total amount of 22,500 m³ of water*" [52] (p. 149); "*total number of 16,017 events . . . b-value of 0.83*" [52] (p. 150). |
| 05pa | Paralana | STIMULATION, "*about $3.1 \times 10^6$ l of water were injected over a period of 5 days . . . 7085 induced microearthquakes detected and located . . . Moment magnitudes range from −0.6 to 2.5, with 90% being larger than 0*" [53] (p. 124); "*The b-value . . . is $1.32 \pm 0.02$ . . . The magnitude of completeness of the catalog is 0.1*" [53] (p. 125). |
| 05sf | Soultz-sous-Forêts | INJECTION, "*1324 triggers were detected and only 449 located events were selected . . . the M of the events ranges from −0.3 to 2.7*" [51] (p. 51); "*lasted about 4 days . . . 30 L/s for 24 h, 45 L/s for 48 h and 25 L/s for 24 h. A total of 12,300 m³ were injected*" [51] (p. 52); b = 0.89 [26] (Table 2). |
| 06ba | Basel | INJECTION, "*Time = 5.5 days, Volume (m³) = 10,800, Event number = 2313, Mi = 1.0, b = 1.65, $\Sigma = 0.4 \pm 0.1$*" [10] (Table 1); "*Max $M_L$ = 3.4*" [23] (Table 1). |
| 07ge | Geysers | INJECTION, "*Between November 2007 and August 2014 about 10.5 Mm3 of treated wastewater was injected . . . 1776 seismic events recorded over the period of nearly 7 years . . . The magnitude of completeness of the resulting catalog is about $M^c{}_W = 1.4$ ($M^c{}_D = 1.0$) and the largest seismic event in the analyzed cluster displayed a magnitude of $M^{max,obs}{}_W = 3.2$*" [54] (p. 7088); "*The average b value is $b = 1.22 \pm 0.08$*" [54] (p. 7093). |
| 07gs | Gross Schönebeck | HYDRAULIC STIMULATION, "*A total of 80 very small ($-1.8 < M_w < -1.0$) induced seismic events were detected*" [45] (p. 995); "*injection was performed over a period of 6 days . . . A total amount of 13,000 m³ of water was injected*" [45] (p. 1000); b = 4.14 [26] (Table 2) not included, as estimated from less than 30 events. |
| 09hn | Hannover | HYDRAULIC STIMULATION, "*Microseismic (1.8 M)*" [24] (Table 1). |
| 10jo | Jolokia | HYDRAULIC STIMULATION, "*injecting a total volume of approximately 380 m³ . . . injection rates were in the order of 1 L/s only*" [55] (p. 199); "*During the 8-day stimulation period, a total of 73 events were detected . . . range between $M_L$ −1.4 and 1.0. Another 139 events occurred within the following six months . . .*" [55] (p. 200). |

**Table A1.** *Cont.*

| ID | Site | Quotes |
|---|---|---|
| 11dp | Desert Peak | STIMULATION TREATMENT, "*a great number of micro-earthquakes (MEQs) with magnitudes ranging from −1.0 to +1.5 were recorded ... the April 2011 stimulation phase is used for analysis*" [56] (p. 140); $0.0 \leq m \leq 0.7$ for Apr. 2011 phase [56] (Figure 1); "*During the April 2011 medium flow-rate phase, ~15 events (out of 18) are located in the vicinity of the STF*" [56] (p. 145); $\dot{V}_{mean} \approx 33$ kg/s and 23 days duration [56] (Figure 5). NB: Many more events induced in Oct–Nov 2011 but no number given; see [56] (Figure 1). |
| 11ma | Mauerstetten | HYDRAULIC STIMULATION [24] (Table 1). |
| 12ha | Habanero | HYDRAULIC STIMULATION, "*injecting a total quantity of 34,000 m³ of water*" [55] (p. 202); "*During the 17.5-day stimulation period, a total of 23,960 seismic events were detected ... range between $M_L$ −1.6 and 3.0. Another 5226 events occurred within the following 30 days ...* " [55] (p. 203); $b = 1.01$ [26] (Table 2). |
| 12nb | Newberry | STIMULATION, "*Over 40,000m³ of ground water was injected*" [57]; "$m_0 = 0.2, b = 0.80, a_{fb} = −2.80$" [13] (Table 2). |
| 13bh | Brady Hot Springs | STIMULATION, "*Early in 2013, a hydraulic stimulation is planned in well 15-12ST1 to extend the reservoir*" [58] (p. 2). NB: Seismic events plotted (magnitude versus time) for >6 experiments between 2011 and 2017 in [59] (p. 16). |
| 13rr | Raft River | THERMAL & HYDRAULIC STIMULATION, "*initiated in June 2013 ... As of August 2014, nearly 90 million gallons have been injected*" [60] (p. 1279); "*Since August 2010, 187 locally generated earthquakes between magnitudes of −1.25 and +1.01 have been measured*" [60] (p. 1281). |
| 13ri | Rittershoffen | HYDRAULIC STIMULATION, "*Approximately 3180 m³ of brine were injected within approximately 21.7 h*" [61] (p. 15); $N = 831, −1.4 \leq m \leq 0.9$ [61] (Table 2); "*During the thermal stimulation and hydraulic stimulation, the b-values were estimated to 1.53 ± 0.15 and 1.16 ± 0.05, respectively*" [61] (p. 21). |
| 13sg | St. Gallen | HYDRAULIC STIMULATION, "*an initial hydraulic stimulation test was performed on 14 July 2013, in which 175 m³ cold water was injected ... On 17 July, two acid stimulations were performed, each injecting 145 m³ ... methane entered the borehole. During the following well-control measures, the operators decided to pump drilling mud (about 700 m³ over 18 h) into the well in order to reduce the pressure buildup*" [62] (p. 7275); "*864 seismic events were detected between 2013 July 14 and December 18. Of these events 349 were strong enough to be located*" [63] (p. 1025); $−1.1 \leq ML(corr) \leq 3.5$ [63] (Figure 6). NB: Important role of gas kick overpressure; $b = 0.83$ [26] (Table 2). |
| 14nb | Newberry | RESTIMULATION, "*2.5 million gallons (9500m³) of groundwater were injected ... 398 events, ranging from M 0 to M 2.26*" [64] (p. 1); "*b-value = 1.01*" [64] (Figure 9); "NB14a $m_0 = 0.0, b = 0.98, a_{fb} = −1.60$, NB14b $m_0 = 0.2, b = 1.05, a_{fb} = −1.60$" [13] (Table 2). |
| 15as | Äspö | STIMULATION, "*A total of 196 picoseismic events were detected*" [65] (p. 6620); "*The estimated moment magnitudes of the AE events ranged between $M_W$ −4.2 ± 0.3 and −3.5 ± 0.3*" [65] (p. 6622); "$b = 2.9 ± 0.2$" [65] (p. 6623); "*The total volume injected into the rock mass during all stimulation phases is approximately 125 dm³*" [65] (p. 6629); $\dot{V}_{mean} \approx 0.05$ L/s [65] (Figure 2). |
| 15re | Reykjanes | HYDRAULIC STIMULATION, "*A sequence of 33 seismic events was observed ... occurred during the stimulation phase ... The moment magnitudes vary from 1.5 to 2.5*" [66] (p. 14); "*high b-value (1.47)*" [66] (p. 9). NB: *V* missing. |
| 16po | Pohang | HYDRAULIC STIMULATION, "*four phases of injection with a total volume of 12,800 m³ at injection rates of 1,00 to 46.83 l s⁻¹*" [67] (p. 1007); $−1.0 \leq m \leq 3.3$, "*log10(N) = 1.97–0.65M*" [68] (Figure 5); "*The seismogenic indices ... extracted from the obtained a values and cumulative injection volumes for A1, A2, and A3 were estimated at −1.69, −1.89, and −1.65, respectively*" [68] (p. 13,068). NB: $m = 5.5$ event not considered as clearly outside the stimulation period. |
| 17gr | Grimsel | HYDRAULIC STIMULATION, "*the higher Mc of −4.32 was used*" [69] (p. 642); $V = 1.4$ m³, $m_{max} = −2.5$ [69] (Figure 1); $N = 65$ [69] (Figure 5); "*b-values of injections into S3 shear zones (HS4: 1.36 ± 0.04; HS5: 1.03 ± 0.05) with highest seismogenic indices (HS4: −3.0; HS5: −2.4)*" [69] (p. 643). NB: Sub-injection HS5 selected as principal experiment as it "*represents the highest-magnitude event as well as the largest seismically activated area*" [69] (p. 643). |

**Table A1.** *Cont.*

| ID | Site | Quotes |
|---|---|---|
| 17re | Reykjanes | STIMULATION, "*Seismic activity was closely monitored during IDDP-2 drilling from the 12th of August 2016 to the 25th of January 2017. During that period 650 earthquakes occurred in the field and more than 200 of them were located within less than 1 km of the IDDP-2 wellhead. The seismic catalogue, however, covering the timespan from the start of drilling to the end of the main stimulation phase that followed the drilling contains over 2300 earthquakes . . . These induced earthquakes were predominantly small, with magnitudes ranging from 0.5 to 1.9 $M_L$*" [70] (p. 5). NB: *V* missing. |
| 18es | Espoo | STIMULATION, "*A total of 18,160 $m^3$ of fresh water was pumped into crystalline rocks over 49 days . . . flow rates of 400 and 800 L/min*" [71] (p. 1); "*the maximum induced event was $M_W$ 1.9*" [71] (p. 2); "*enlarged the original near-real-time industrial seismic catalog to 43,882 events, with magnitudes down to $M_W = -0.6$*" [71] (p. 5); "*the b value returned to and then remained at ~1.3*" [71] (p. 6). |
| 19ve | Vendenheim | 11 earthquakes within a month with $m_{max}$ = 3.5 [72]. |

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
