# Peer review of "Comprehensive Survey of Seismic Hazard at Geothermal Sites by a Meta-Analysis of the Underground Feedback Activation Parameter afb"

_energies, doi:10.3390/en14237998_

Round 1

Reviewer 1 Report

First of all, the manuscript needs to transform to energies template, and also the reference needs to be further processed with the MDPI energies authors guidelines.

For the manuscript, the aim of the study is clear as the authors made a meta-analysis of the underground feedback activation parameter as a comprehensive survey of seismic hazards at several geothermal sites. The references are recent and relevant. The introduction is clear about what is already known.

Then the authors make a literature mining and a meta-analysis and then introduce the results.

My objection is the section Discussion, please split the section to Discussion and Conclusions.

In general, a very interesting manuscript that I want to review again in full template format and clear discussion and conclusion section.   

Also I attach a pdf with some typos

Author Response

Thank you for your comments and recommendations. We have now reformatted the manuscript following the Energies journal template and Energies reference system. We also split the Discussion section into Discussion and Conclusions. We modified the text accordingly. Finally, we corrected the typos mentioned in the PDF file.

Reviewer 2 Report

There are 66 references in the list of references and there is one without a date (line 294). Considering the ones with dates, about 37% are less than 5 years old, and about 37% are more than 10 years old. The average date is 2009.

I found the paper interesting. However, it is not identified in the manuscript the type of paper, namely if it is a review or a research paper, because it seems to be a sort of a systematic revision (if so, it is incomplete in my opinion) with a meta-analysis, but the goal is not clearly stated in the paper, namely in the abstract. Moreover, in my opinion, in any case the authors should provide a more detailed introduction about induced seismicity  (including important factors like the permeability along the fracture path or the role of initial stress state, because many researchers stated that the initial stress controls the nature of the fluid induced seismicity), namely adding some recent references about the phenomenon so that less specialized readers can deepen their knowledge of the subject.

For example:

Rathnaweera, T.D.; Wu, W.; Ji, Y.; Gamage, R.P. Understanding injection-induced seismicity in enhanced geothermal systems: From the coupled thermo-hydro-mechanical-chemical process to anthropogenic earthquake prediction. Earth-Science Reviews 2020, 205, 103182, doi: https://doi.org/10.1016/j.earscirev.2020.103182.

Deng, Q.; Blöcher, G.; Cacace, M.; Schmittbuhl, J. Modeling of fluid-induced seismicity during injection and after shut-in. Computers and Geotechnics 2021, 140, 104489, doi: https://doi.org/10.1016/j.compgeo.2021.104489.

The authors should provide a reason for using b=1.16 (line 98) as the mean value. Because the fractal dimension is 2 x b, if the earthquakes fill a plane, it should be b=1. If the earthquakes fill a volume, it should be b=1.5 (which is more related to volcanic seismicity, namely when associated to eruptions).

I believe that the paper is interesting for publishing, but only after a revision.

Author Response

Thank you for your comments and recommendations. The type of paper is now indicated as ‘Article’. The goal is now clearly stated in the abstract: We followed a two-step procedure with a review of the literature followed by a meta-analysis. We observed the gaps in the literature concerning the critical parameter afb and therefore extracted from articles and reports other parameters from which we could infer afb. This leads to the meta-analysis and a new afb distribution (that is comprehensive).

We additionally extended the introduction by providing more details about the physics of induced seismicity (where we added the 2 mentioned references and one more from Shapiro’s group).

Regarding the b-value, we followed a purely data-driven approach and hence chose the expectation, or mean, as the most reasonable value. Even if b = 1 is commonly observed in tectonic regimes and 1.5 in volcanic regimes, induced seismicity can occur in an isotropic cloud or along existing fault planes. As such, the mean 1.2 appears as a good compromise between the two possible regimes. Finally imputation by the mean is a frequently used approach in data science.

Round 2

Reviewer 1 Report

Dear Authors, I saw that you address all my comments.

Good work!

With regards 

The reviewer

Reviewer 2 Report

The authors have slightly improved the introduction, and clarified that the manuscript is a research paper, responding to my concerns.

They also presented arguments about the choice of the mean value adopted for the parameter b of the Gutenberg-Richter law.

Even though I believe that the article could have been greatly improved, it already presents the minimum conditions to be published.